# Relationship among Left Ventricular Hypertrophy, Cardiovascular Events, and Preferred Blood Pressure Measurement Timing in Hemodialysis Patients

**DOI:** 10.3390/jcm9113512

**Published:** 2020-10-30

**Authors:** Hiroaki Io, Junichiro Nakata, Hiroyuki Inoshita, Masanori Ishizaka, Yasuhiko Tomino, Yusuke Suzuki

**Affiliations:** 1Department of Nephrology, Juntendo University Nerima Hospital, Tokyo 177-8521, Japan; ino-hi@juntendo.ac.jp (H.I.); m-ishiza@juntendo.ac.jp (M.I.); 2Department of Nephrology, Juntendo University Faculty of Medicine, Tokyo 113-8412, Japan; jnakata@juntendo.ac.jp (J.N.); yasu@mtnet.jp (Y.T.); yusuke@juntendo.ac.jp (Y.S.)

**Keywords:** left ventricular hypertrophy, left ventricular mass index, hemodialysis, blood pressure, cardiovascular events, risk factor

## Abstract

This study aimed to identify the ideal timing and setting for measuring blood pressure (BP) and determine whether the left ventricular mass index (LVMI) is an independent risk factor associated with increased cardiovascular events in hemodialysis (HD) patients. BP and LVMI were measured at baseline and at 6 and 12 months after HD initiation. BP was monitored and recorded at nine different time points, including before and after HD over a one-week period (HDBP). The mean BP measurement was calculated as the weekly averaged BP (WABP). LVMI was significantly correlated with home BP, in-office BP, HDBP, and WABP. Receiver operating characteristic analysis indicated that the cutoff LVMI value for cardiovascular events was 156 g/m^2^. LVMI and diabetes mellitus were significant influencing factors for cardiovascular events (hazards ratio (95% confidence interval): diabetes mellitus, 2.84 (1.17,7.45); LVMI > 156 g/m^2^, 2.86 (1.22,6.99)). Pre-HDBP, post-HDBP, and WABP were independently associated with higher LVMI in the follow-up periods. Hemoglobin and human atrial natriuretic peptide (hANP) levels were associated with LVMI beyond 12 months after HD initiation. Treatment of hypertension, overhydration based on hANP, and anemia may reduce the progression of LVMI and help identify HD patients at high risk for cardiovascular events.

## 1. Introduction

Cardiovascular (CV) disease is a leading cause of morbidity and mortality in hemodialysis (HD) patients. The US Renal Data System 2014 Annual Data Report stated that, between 2011 and 2013, cardiac arrest was the primary cause of two-thirds of CV-related deaths in HD patients [1]. In this population of patients, both volume and pressure overload result in increased cardiac work. The improvements in the prevention or postponement of kidney failure in the United States are possibly due to interventions such as greater blood pressure control in the general population. The prevalence of end-stage kidney disease (ESKD) continues to increase and reached 746,557 cases in 2017 (vs. 727,912 in 2016), representing a 2.5% increase since 2016, a reflection of decreasing mortality rates in the ESKD population [2]. The adaptive response to this physiological situation leads to left ventricular hypertrophy (LVH). LVH is a common comorbid condition observed with chronic kidney disease (CKD) and is a significant predictor of increased CV events in dialysis patients [3]. At the initiation of dialysis therapy, the prevalence of LVH is high [4], possibly due to both delayed diagnosis and insufficient treatment of hypertension.

Echocardiography provides an accurate estimate of LV mass (LVM). The diagnosis of LVH using echocardiography is frequently seen in patients with ESKD [5]. In early CKD patients, the LVM index (LVMI), as estimated by echocardiography, has been shown to correlate with the risk of progression to dialysis [6]. Lower blood hemoglobin level and higher LVMI are associated with progression to dialysis and poorer outcomes [7].

The optimal timing and measurement technique of BP in patients with ESKD are yet to be established. Currently, there is no established method of BP measurement that more accurately predicts the development of elevated LVMI and LVH and thus increased risk for CV events [8]. A significant change in BP, through a decrease in LV end-diastolic pressure, might decrease the left atrial (LA) diameter, altering the assessment of the LVH calculation of LVMI on echocardiography [9]. Many studies have used pre-HD BP to determine optimal BP levels in HD patients [10].

Moriya et al. [11] reported that the weekly averaged BP (WABP) is a useful method for estimating the BP of HD patients and correlates well with the LVMI. In HD patients, however, the relationship between LVH and BP variability remains unclear [12]. After starting HD, systolic BP, atrial natriuretic peptide (ANP), and hemoglobin levels have been found to be predictive factors for LVMI [3,9], more so than diastolic BP. Matsumoto et al. [13] reported that an increase in hemoglobin level along with a reduction in ANP were associated with reductions in LVH. Intensive HD is associated with lower risks for cardiovascular death and hospitalization, especially for heart failure, relative to both conventional HD and peritoneal dialysis [14]. In Japan, the incidence of ESKD has increased in the last few decades [15]. Although studies have reported an association between the prevalence of LVH and CV risk factors in patients with CKD [16], the relationship between LVMI and CV events in HD patients requires further exploration. Thus, this study aimed to assess when and how BP should be monitored in relation to HD and whether LVMI is an independent risk factor for increased CV events in HD patients.

## 2. Materials and Methods

The study protocol was approved by the Ethics Review Committee of Juntendo University Faculty of Medicine, Tokyo, Japan, in 2007 (approval no. 207-036, dated 16 October 2007) and registered with the University Hospital Medical Information Network (UMIN000018312). The modified study design (prospective observational study) was approved in 2015 (approval no. 22-78; 2015026). It complied with the tenets of the 2000 Declaration of Helsinki.

This prospective observational study was conducted in the Department of Nephrology of the Juntendo University Hospital. The inclusion criteria for the study were adults aged >20 years who were newly indicated for HD. Patients who were unable to revisit the outpatient clinic 6 months after initiating HD and those with concomitant malignancy, alcoholism, or chronic inflammatory disease were excluded.

Demographic data were collected at baseline. We measured the patients’ human ANP (hANP), hemoglobin, high-sensitivity C-reactive protein (hsCRP), albumin, homocysteine, iron, and transferrin saturation levels at baseline (day 0), and at 6 and 12 months after initiation of HD. Of the 418 HD patients, 192 provided consent. A total of 72 patients were excluded from the study (22 had malignancy, 2 had alcohol abuse, 31 had inflammation or infection, 14 could not come to the hospital 6 months later due to relocation or hospital transfer, and 3 withdrew their consent), and the remaining 120 patients who survived beyond 6 months were included in this study.

### 2.1. BP Measurement

We measured and recorded 9 BP measurements over a one-week period for each patient. Home BP (HBP) was measured in the mornings on treatment days 1 and 5 of each week. BP was measured just before HD on each dialysis day (days 1, 3, and 5) (pre-HDBP) and again after each dialysis (post-HDBP). Finally, BP was measured once during their clinic echocardiogram visit day (day 4 of each week) (visit BP(VBP)). The WABP was defined as the average of these 9 BP measurements (Figure 1). Nine BP measurements were performed at baseline and at 6 and 12 months.

### 2.2. Echocardiography Measurement

Echocardiographic examinations were conducted on a non-HD day. In all patients, echocardiographic examinations were performed by one examiner using the Toshiba ultrasound system model 260 SS-A equipped with a 2.5-MHz phased-array transducer (Toshiba, Tokyo, Japan). All examinations were performed with the patient lying in the left lateral recumbent position. Data were analyzed following the American Society of Echocardiography Guidelines. The LA and LV sizes, interventricular septal thickness, and LV posterior wall thickness were measured by two-dimensional and M-mode echocardiography [17]. We measured the LVMI [18], early/late LV filling velocity (E/A), and deceleration time. The LV mass was calculated using the following formula [19]:The LV mass = 0.8 (1.04 (LV internal diameter in diastole + posterior wall thickness + interventricular septal thickness)^3^ – (LV internal diameter in diastole)^3^) + 0.6 g(1)

LVMI was defined as LV mass standardized by the body surface area [19].

### 2.3. CV Events

A CV event was defined as hospitalization for unstable angina, CV death, sustained arrhythmia, myocardial infarction, stroke, arteriosclerosis obliterans, transient ischemic attack, arterial aneurysm, and valve disease. Admission for fluid overload and pulmonary edema was included in those with a diagnosis of angina or myocardial infarction. Mere fluid overload due to insufficient dry weight settings was not included in the CV event.

### 2.4. Statistical Analysis

Statistical analysis was performed using JMP 10 (SAS Institute, Cary, NC, USA). Standard descriptive statistics were used to assess baseline characteristics. Data are presented as the mean ± standard deviation. Characteristics of patients with and without CV events were compared using analysis of variance (ANOVA). Variables with *p* values < 0.05 were further analyzed using a stepwise linear regression analysis based on a forward–backward procedure. Repeated-measures ANOVA was performed to compare serial changes in the clinical data and echocardiographic parameters. Cox regression was used to analyze relative CV event risks. We chose hANP as a measure of capacity load and BP as a measure of pressure load associated with LVH, in addition to LVMI, age, sex, diabetes mellitus (DM), BP, and LVMI, to evaluate the association between LVMI and CV events. The independence of these variables was evaluated to select variables to be used for multivariate Cox hazard analysis. A cutoff value of LVMI was calculated for CV events using receiver operating characteristic (ROC) analysis based on the value at 6–12 months after the start of dialysis, creating a binary categorical variable based on LVMI. The cumulative CV event incidence was assessed using the Kaplan–Meier method and log-rank test using significant influencing factors for CV events based on the multivariate Cox hazard model. Multivariate linear regression analysis was used to examine the association between LVMI or hemoglobin and clinical and laboratory variables. A value <0.05 was considered statistically significant.

## 3. Results

### 3.1. Patient Characteristics

A total of 418 patients who were admitted to our hospital between October 2008 and September 2012 for HD were screened. After applying the exclusion criteria, the final study population included 120 HD patients. The mean patient age was 63.5 ± 11.4 years at the initiation of HD. Most patients were male (73.3%), 42.6% had diabetes, and 30.8% had hypertensive nephrosclerosis. Diabetic kidney disease was defined as a patient with a history of diabetes for more than 5–10 years and diabetic retinopathy and/or high proteinuria.

### 3.2. Baseline and 12 Months Echocardiographic, Laboratory, and BP Values

Hemoglobin (g/dL), hANP (pg/mL), and hsCRP (mg/dL) levels differed significantly after 12 months of HD. In contrast, the LVMI at month 12 of HD (145.5 ± 46.2 g/m^2^) was not significantly different from that at month 0 (178.1 ± 48.5 g/m^2^). Only the end-of-week post-HD systolic BP decreased significantly after 12 months of HD (157.6 ± 18.5 vs. 136.8 ± 19.9 mmHg). Systolic and diastolic VBP, home systolic and diastolic BP (HBP), pre- and post-HD systolic BP, post-HD diastolic BP, and WABP did not change significantly at 12 months post-HD initiation.

### 3.3. Characteristics of Patients with and without CV Events 12 Months after Initiation of HD

Table 1 shows the differences between patients with and without CV events. There were no deaths. CV events included angina pectoris (*n* = 12), cerebral infarction (*n* = 6), occlusive arterial disease (*n* = 2), myocardial infarction (*n* = 1), and arrhythmia (*n* = 2). Age >70.6 years, DM, past CV event, and elevated hsCRP levels, were all significantly correlated with an elevated risk of CV events.

### 3.4. Cox Proportional Hazards Modeling

As a result of the ROC analysis, the calculated cutoff value of LVMI for CV events was 156 g/m^2^. LVMI, age, DM, sex, hANP, and BP (outpatient systolic BP at 6–12 months) were considered as candidates independent variables in the examination of the influence of LVMI on CV events. Evaluation of the independence of these variables showed that LVMI was significantly correlated with hANP and BP (outpatient systolic BP). In addition, hANP, BP, and age were all significantly different between the two LVMI groups. Since the number of events in this instance was 23, the independent variable that can be input to the Cox proportional hazards model was approximately 1/10 of the events, which was approximately 2 to 3. The current variable was selected because only approximately three independent variables were available. BP, hANP, and age were excluded from the above six variable candidates, and analysis using a multivariate Cox hazard model was performed using LVMI, sex, and DM as independent variables. As a result, the model showed significance (*p* = 0.0068): LVMI and DM were found to be significant influencing factors for CV events (hazards ratio (95% confidence interval): DM, 2.84 (1.17–7.45); LVMI >156 g/m^2^, 2.86 (1.22–6.99)) (Table 2).

A significant difference was found in the cumulative CV event incidence rates among the four subgroups classified according to the LVMI classification and the presence or absence of DM, both of which were significant influencing factors in the Cox hazard model (*p* = 0.0059) (Figure 2).

Sex, BP, and hANP were the factors not related to CV events that were considered in the univariate analysis. All patients who were included in the analysis were taking angiotensin receptor blockers or angiotensin-converting enzyme inhibitors. Furthermore, the analysis of the cardiothoracic ratio (CTR), a simple method for evaluating dry weight in dialysis patients of Japanese origin, was not possible in our cohort because the CTR data included both pre- and post-dialysis values.

We presented the baseline characteristic with the patients grouped according to LVMI <156 g/m^2^ and LVMI >156 g/m^2^ (Table 3).

### 3.5. Factors Associated with LVMI 12 Months after Initiation of HD

The univariate regression analysis showed that LVMI was significantly correlated with start- and end-of-week waking systolic HBP, systolic VBP, pre-HD systolic BP, post-HD systolic BP, WABP, and hemoglobin levels at 12 months after HD initiation (Table 4). hANP was independently associated with LVMI after 6 months and 12 months (Table 4). Hemoglobin levels were significantly correlated with end-of-week post-HD systolic BP. LVMI was positively correlated with systolic blood pressure at almost all blood pressure measurements. Sex, LVMI (binary variable classified by 156 g/m^2^), and blood pressure were selected as independent variables, and blood pressure was evaluated in each model using VBP, WABP, start of week before HD-BP, and end of week after HD-BP. For significant models, the *p* value for each variable and the odds ratio [95% confidence interval] to past CV events were calculated. In model 4, high LVMI and end of week after HD-BP were found to be associated with past CV events. In other words, high LVMI showed an increase in the incidence of past CV events, and low blood pressure after weekend dialysis showed an increase in past CV events (Table 5).

## 4. Discussion

In this prospective observational study, we found an association between increased CV events and LVMI > 156 g/m^2^ with DM after performing multivariate regression analysis. End-of-week post-HD systolic BP significantly decreased 12 months after HD initiation. We also found that pre-HDBP at the start of the week, post-HDBP at the end of the week, and WABP were independently associated with LVMI on univariate regression analysis of follow-up. Multiple BP measurements taken before and after dialysis and during dialysis were reconfirmed to be the most accurate assessment format. With regard to monitoring BP, the time point at which the BP is measured should be defined clearly. Moreover, because the BP of HD patients varies with each HD session as a result of loss of excess fluid, BP fluctuation should be considered. It is important not to limit BP evaluation to only one measurement, such as before dialysis or after dialysis; rather, multiple measurements of BP should be performed and then averaged. In this study, the number of DM cases was significantly higher in patients with CVE than in those without CVE. VBP was significantly higher in patients with LVMI > 156 g/m^2^ than in those with LVMI < 156 g/m^2^. The results of this study indicated that patients with low blood pressure after weekend dialysis showed an increase in past CV events. A previous study reported that systolic blood pressure < 110 mmHg and DBP < 70 mmHg were independent risk factors of CV events and all-cause mortality [20], which supports our findings. A recent study that performed sensitivity analysis using LVH and RWT separately showed that LVH but not RWT was associated with higher cardiorenal risk [21]. In this study, LVMI and RWT were not significantly different between patients with CV and those without CV events. From the pathophysiologic standpoint, an increase in afterload (i.e., arterial hypertension or an increase in large arteries stiffness) can induce concentric LVH, whereas volume overload (i.e., anemia and hypervolemic states) leads to eccentric LVH [22].

We observed that hemoglobin levels differed at 12 months after initiating HD, but hemoglobin was not an independently associated factor with LVMI. A further analysis indicated that hemoglobin levels were correlated with end-of-week post-HD systolic BP. A previous study on the independent effect of BMI reported a greater effect on LVH in women than in men [23]. There are no new discoveries about LVMI and BMI in this study.

In our study, albumin was one of the factors associated with LVMI 12 months after HD initiation. This finding is consistent with other studies showing that serum albumin is negatively correlated with LVMI [18]. hsCRP, a marker of inflammation, is also correlated with a high rate of major CV events and has been described as an independent predictor of LVMI in patients with CKD [24]. The present findings support previous observations.

Our study has some limitations. First, it was not double-blinded, and there was no control group in this study. Second, as an observational study, the findings established an association rather than a causal relationship among systolic BP, hemoglobin, and LVH. Third, the study did not evaluate patients’ compliance with home ambulatory BP monitoring or compliance with antihypertensive medication regimens. Furthermore, we did not apply highly precise and reliable cardiac magnetic resonance imaging, the new gold standard for measuring LVMI [25].

Nevertheless, the findings of our study clearly defined the relative role of LVM with diabetic kidney disease in the risk assessment of patients with ESKD. Echocardiography may aid in the identification of a group at higher risk of developing CV events in HD patients. While assessing LVMI after the initiation of HD may be difficult, the treatment of hypertension, overhydration based on hANP, and anemia after dialysis initiation may reduce the progression of LVH and help identify patients at high risk for cardiovascular events before and after HD initiation.

## Figures and Tables

**Figure 1 jcm-09-03512-f001:**
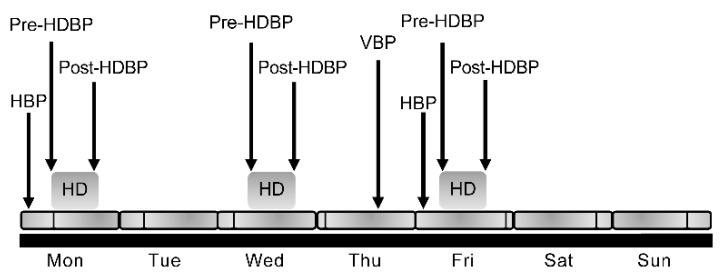
Blood pressure measurements over a one-week period. HBP, home blood pressure; VBP, visit blood pressure; HD; hemodialysis; HDBP, hemodialysis blood pressure.

**Figure 2 jcm-09-03512-f002:**
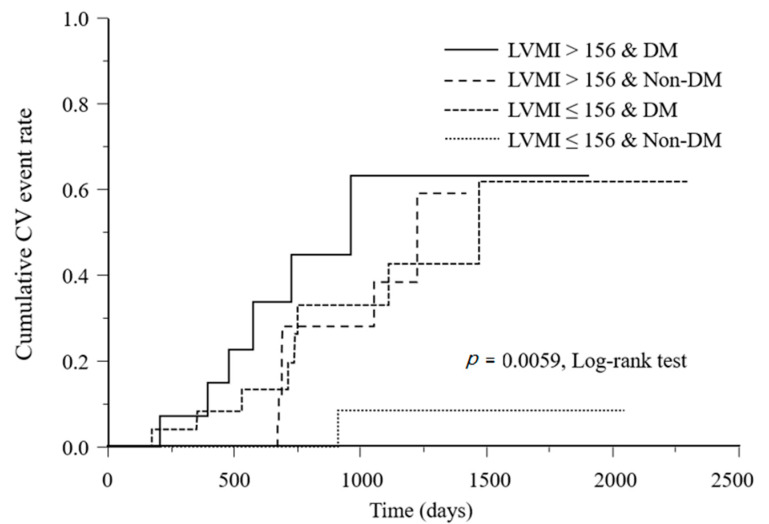
Kaplan–Meier analysis of CV event incidence according to LVMI class and DM. CV, cardiovascular; LVMI, left ventricular mass index; DM, diabetes mellitus.

**Table 1 jcm-09-03512-t001:** ANOVA of characteristics between patients with and without CV events 12 months after initiation of HD.

	Patients with CV Events	Patients without CV Events	*p* Value	All Patients
Age (years)	70.6 ± 2.4	61.3 ± 1.4	<0.01	63.5 ± 11.4
Male sex (%)	77.3	72.5	0.65	73.6
DM (%)	63.6	36.1	0.02	42.6
Past CV event (%)	60.1	18.3	<0.01	27.7
Systolic blood pressure (mmHg)				
Outpatient clinic	135.2 ± 4.4	142.6 ± 2.3	0.15	140.9 ± 1.8
Before hemodialysis (start of the week)	148.2 ± 5.5	147.2 ± 3.1	0.87	147.4 ± 2.1
Mean blood pressure	143.1 ± 3.5	141.1 ± 3.5	0.63	141.6 ± 1.1
Laboratory parameters				
Hemoglobin (g/dL)	10.9 ± 0.3	10.6 ± 0.1	0.24	10.7 ± 0.1
Non-dialysis day hemoglobin (g/dL)	11.5 ± 0.4	11.5 ± 0.4	0.87	11.5 ± 0.2
Total iron saturation (%)	22.1 ± 3.5	27.9 ± 23.8	0.15	26.4 ± 14.7
Albumin (g/dL)	3.79 ± 0.07	3.77 ± 0.04	0.84	3.77 ± 0.03
Homocysteine (mg/dL)	23.7 ± 6.2	34.7 ± 3.1	0.11	32.7 ± 2.2
hsCRP (mg/dL)	0.49 ± 0.09	0.13 ± 0.05	<0.01	0.21 ± 0.04
hANP (pg/ dL)	68.4 ± 12.5	60.5 ± 6.3	0.58	62.1 ± 4.9
Alkaline phosphatase (U/L)	258.1 ± 18.7	232.1 ± 10.2	0.23	238.1 ± 8.2
Corrected calcium (mg/dL)	8.69 ± 0.28	8.97 ± 0.16	0.39	8.89 ± 0.12
Phosphorous (mg/dL)	5.31 ± 0.29	5.39 ± 0.16	0.74	5.38 ± 0.13
Total cholesterol (mg/dL)	158.3 ± 8.2	168.3 ± 4.7	0.29	165.8 ± 3.7
LDL cholesterol (mg/dL)	84.6 ± 7.9	90.9 ± 4.3	0.49	89.4 ± 3.4
Intact PTH (pg/dL)	171.1 ± 29.4	152.8 ± 15.4	0.58	156.7 ± 12.1
Erythropoietin stimulating agent (U/week)	21.1 ± 3.2	18.9 ± 1.9	0.54	19.4 ± 1.4
Echocardiographic data				
LAD (mm)	37.7 ± 1.3	35.8 ± 0.7	0.21	36.3 ± 0.6
Relative wall thickness	0.528 ± 0.02	0.496 ± 0.01	0.16	0.503 ± 0.01
LVMI (g/m^2^)	155.4 ± 7.9	144.6 ± 4.2	0.23	146.9 ± 3.4
EF (%)	66.8 ± 2.3	68.2 ± 1.2	0.61	67.9 ± 0.9
Follow-up period (days)	532.4 ± 104.2	728.1 ± 57.2	0.11	682.8 ± 48.8

ANOVA, analysis of variance; CV, cardiovascular; DM, diabetes mellitus; HD, hemodialysis; hsCRP, high-sensitivity C-reactive protein; hANP, human atrial natriuretic peptide; LDL, low-density lipoprotein; PTH, parathyroid hormone; LAD, left atrial dimension; LVMI, left ventricular mass index; EF, ejection fraction.

**Table 2 jcm-09-03512-t002:** Adjusted hazard ratio for CV events.

Variables	HR (95% CI)	*p* Value
Male sex	1.44 (0.45,6.41)	0.5595
DM	2.84 (1.17,7.45)	0.0214
LVMI > 156 g/m^2^	2.86 (1.22,6.99)	0.0154

Multivariate Cox hazard model: *p* = 0.0068. CV, cardiovascular; CI, confidence interval; DM, diabetes mellitus; HR, hazard ratio; LVMI, left ventricular mass index.

**Table 3 jcm-09-03512-t003:** Comparison of baseline characteristics between patients with LVMI < 156 g/m^2^ and patients with LVMI > 156 g/m^2^.

	LVMI < 156 g/m^2^	LVMI > 156 g/m^2^	*p* Value	All Patients
Age (years)	61.4 ± 13.2	63.7 ± 11.9	0.37	63.0 ± 11.4
Male sex (%)	67.5	73.7	0.51	71.8
DM (%)	41.2	46.8	0.58	45.1
Systolic blood pressure (mmHg)				
Outpatient clinic	139.8 ± 18.5	147.7 ± 18.2	<0.05	144.8 ± 18.4
Before hemodialysis (start of the week)	147.2 ± 26.5	147.6 ± 19.5	0.95	147.5 ± 22.1
Mean blood pressure	142.9 ± 18.9	140.5 ± 13.8	0.63	141.3 ± 15.1
Labolatory parameters				
Hemoglobin (g/dL)	9.1 ± 1.1	8.8 ± 1.1	0.36	8.9 ± 1.1
TSat (%)	21.7 ± 8.8	28.2 ± 16.4	0.11	26.2 ± 14.6
Albumin (g/dL)	3.75 ± 0.37	3.77 ± 0.28	0.84	3.77 ± 0.32
Homocysteine (mg/dL)	37.8 ± 28.8	31.6 ± 20.2	0.32	33.6 ± 22.2
hsCRP (mg/dL)	0.18 ± 0.08	0.24 ± 0.06	0.59	0.22 ± 0.06
hANP (pg/dL)	50.9 ± 29.6	65.4 ± 57.1	0.24	60.6 ± 44.9
Alkaline phosphatase (U/L)	256.2 ± 70.7	230.3 ± 85.1	0.18	239.4 ± 81.2
Phosphorous (mg/dL)	5.3 ± 0.8	5.3 ± 1.4	0.89	5.3 ± 1.3
Calcium-phosphorous product (mg^2^/dL^2^)	47.5 ± 8.3	48.3 ± 12.9	0.77	48.1 ± 11.2
Total cholesterol (mg/dL)	168.5 ± 41.1	160.9 ± 30.1	0.36	163.4 ± 35.7
LDL cholesterol (mg/dL)	91.1 ± 42.1	88.3 ± 27.1	0.73	89.2 ± 33.1
Intact PTH (pg/dL)	185.4 ± 23.9	137.6 ± 16.8	0.11	153.4 ± 18.1

LVMI, left ventricular mass index; DM, diabetes mellitus; TSat, transferrin saturation; hsCRP, high-sensitivity C-reactive protein; hANP, human atrial natriuretic peptide; LDL, low-density lipoprotein; PTH, parathyroid hormone.

**Table 4 jcm-09-03512-t004:** Univariate analysis of factors associated with LVMI beyond 12 months after HD initiation.

Factor	*r* Value	*p* Value
Hemoglobin (HD day)	−0.154	<0.001 *
Hemoglobin (non-HD day)	−0.118	0.08
hANP	0.514	<0.0001 *
hsCRP	0.112	0.083
Homocysteine	−0.164	0.019 *
Albumin	−0.176	0.0017 *
Iron	−0.111	0.04 *
TSat	−0.107	0.22
Systolic VBP	0.287	<0.0001 *
Diastolic VBP	0.011	0.866
Start-of-week waking systolic HBP	0.256	0.0015 *
Start-of-week waking diastolic HBP	0.087	0.448
End-of-week waking systolic HBP	0.341	<0.0001 *
End-of-week waking diastolic HBP	0.087	0.206
Start-of-week pre-HD systolic BP	0.289	<0.0001 *
Start-of-week pre-HD diastolic HDBP	0.087	0.291
End-of-week post-HD systolic HDBP	0.240	<0.0001 *
End-of-week post-HD diastolic HDBP	−0.005	0.948
WABP	0.311	<0.0001 *

LVMI, left ventricular mass index; HD, hemodialysis; hANP, human atrial natriuretic peptide; hsCRP, high-sensitivity C-reactive protein; TSat, transferrin saturation; VBP, blood pressure at the time of visit to the hospital on non-HD day; HBP, home blood pressure; HDBP, blood pressure before or after HD session; WABP, weekly averaged blood pressure. *: *p* < 0.05

**Table 5 jcm-09-03512-t005:** Adjusted hazard ratio for past CV events.

Variables	HR (95% CI)	*p* Value
Model 1		0.0388
Male sex	4.39 (0.88,21.78)	0.0399
LVMI (>156 g/m^2^)	2.86 (0.99,8.19)	0.0479
VBP	1.00 (0.97,1.03) (unit odds)	0.9937
Model 2		0.0718
Male sex	-	-
LVMI (>156 g/m^2^)	-	-
WABP	-	-
Model 3		0.0514
Male sex	-	-
LVMI (>156 g/m^2^)	-	-
Start-of-week pre-HDBP	-	-
Model 4		0.0002
Male sex	6.58 (0.75,57.44)	0.0882
LVMI (>156 g/m^2^)	4.56 (1.17,17.83)	0.0291
End-of-week post-HDBP	0.94 (0.91,0.98) (unit odds)	0.0031

CV, cardiovascular; HR, hazard ratio; CI, confidence interval; LVMI, left ventricular mass index; VBP, blood pressure at the time of visit to the hospital on non-HD day; WABP, weekly averaged blood pressure; HDBP, blood pressure before or after HD session.

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
