# Peer review of "Relationship among Left Ventricular Hypertrophy, Cardiovascular Events, and Preferred Blood Pressure Measurement Timing in Hemodialysis Patients"

_jcm, 2020, doi:10.3390/jcm9113512_

Round 1
Reviewer 1 Report
Good interesting study, see recommendations below.
Need more recent references in intro and background section. Need to quote statistics from USRDS 2019
Should also include findings of article below in background information.
McCullough, P. A., Chan, C. T., Weinhandl, E. D., Burkart, J. M., & Bakris, G. L. (2016). Intensive hemodialysis, left ventricular hypertrophy, and cardiovascular disease. American Journal of Kidney Diseases, 68(5), S5-S14.
Also under methodology to were 9 BP measurements checked at baseline, 6 months and 12 months? Need to clarify this.
Need to be more specific about inclusion/ How did you attain a sample of 120 patients from 418 patients? exclusion criteria was not returning to clinic at 6 months only exclusion criteria?
Need to be more specific as to how you categorized LVMI in narrative section of paper? ie LVMI > 146 or < 146
Statistical analysis is somewhat confusing - under section 3.4 - you mention that LVMI was significantly correlated with hANP and BP, however you excluded these variables from the multivariate model - need to clarify this and have a statistician review again. -
Where are the findings for the multiple regression for section 3.5 - how much of an effect did these variables have on the outcome of LVMI?
Need to incorporate more of the findings in the literature into the discussion.
Author Response
The paper was revised according to the suggestions and comments of the reviewer’s comments. In the revised manuscript, revisions are in red and underlined, and deletions are in red with strikeout. Some new comments and changes in the text were added. We sincerely believe that we have adequately answered all queries and issues. We would like to thank the editor and reviewers for the thorough review of our paper, which helped improve the manuscript.
Reviewer’s 1 comments and point-by-point answers
- Need more recent references in intro and background section. Need to quote statistics from USRDS 2019
Answer: The improvements in the prevention or postponement of kidney failure in the United States were possibly due to interventions such as greater blood pressure control in the general population. The prevalence of ESRD continues to increase and reached 746,557 cases in 2017 (vs 727,912 in 2016), representing a 2.5% increase since 2016, a reflection of decreasing mortality rates in the ESRD population (ref).
Ref: Saran, R.; Li, Y.; Robinson, B.; Ayanian, J.; Balkrishnan, R.; Bragg-Gresham, J.; Chen, J.T.; Cope, E.; Gipson, D.; He, K.; Herman, W.; Heung, M.; Hirth, R.A.; Jacobsen, S.S.; Kalantar-Zadeh, K.; Kovesdy, C.P.; Leichtman, A.B.; Lu, Y.; Molnar, M.Z.; Morgenstern, H.; Nallamothu, B.; O'Hare, A.M.; Pisoni, R.; Plattner, B.; Port, F.K.; Rao, P.; Rhee, C.M.; Schaubel, D.E.; Selewski, D.T.; Shahinian, V.; Sim, J.J.; Song, P.; Streja, E.; Tamura, M.K.; Tentori, F.; Eggers, P.W.; Agodoa, L.Y.C.; Abbott, K.C. US Renal Data System 2014 Annual Data Report: epidemiology of kidney disease in the United States. Am. J. Kidney Dis. 2015, 66, Svii, S1–S305.
These sentences were added to the “Introduction” section.
- Should also include findings of article below in background information.
Answer: Intensive HD is associated with lower risks for cardiovascular death and hospitalization, especially for heart failure, relative to both conventional HD and peritoneal dialysis (ref).
Ref: McCullough, P.A.; Chan, C.T.; Weinhandl, E.D.; Burkart, J.M.; Bakris, G.L. Intensive hemodialysis, left ventricular hypertrophy, and cardiovascular disease. Am. J. Kidney Dis. 2016, 68, S5–S14.
These sentences were added to the “Introduction” section.
- Also under methodology to were 9 BP measurements checked at baseline, 6 months and 12 months? Need to clarify this.
Answer: Nine BP measurements were performed at baseline and at 6 and 12 months.
This sentence is added to the “Methods” section.
- Need to be more specific about inclusion/ How did you attain a sample of 120 patients from 418 patients? exclusion criteria was not returning to clinic at 6 months only exclusion criteria?
Answer: Of the 418 HD patients, 192 provided consented (22 had malignancy, 2 had alcohol abuse, 31 had inflammation or infection, 14 could not come to the hospital 6 months later due to relocation or hospital transfer, and 3 withdrew their consent).
These sentences were added to “Materials and Methods” section.
- Need to be more specific as to how you categorized LVMI in narrative section of paper? ie LVMI > 146 or < 146
Answer: In addition to LVMI as a continuous variable that did not differ significantly between the presence and absence of CV events, we added a categorical variable for LVMI classified by a median of 146 g/m2 that showed a significant difference between groups. However, we believe that it is not appropriate to use anything other than the cutoff value (156 g/m2) calculated by the ROC analysis for the presence or absence of CV events. Therefore, the notation of 146 g/m2 in LVMI in Table 2 was deleted.
- Statistical analysis is somewhat confusing - under section 3.4 - you mention that LVMI was significantly correlated with hANP and BP, however you excluded these variables from the multivariate model - need to clarify this and have a statistician review again.
Answer: Since the number of events in this instance was 23, the independent variable that can be input to the Cox proportional hazards model was approximately 1/10 of the events, which was approximately 2 to 3. The current variable was selected because only approximately 3 independent variables were available
These sentences were added to section 3.4 of the “Results” section.
- Where are the findings for the multiple regression for section 3.5 - how much of an effect did these variables have on the outcome of LVMI?
Answer: Table 4 was a univariate analysis, not a multivariate analysis. We corrected fixed it. LVMI was positively correlated with systolic blood pressure at almost all blood pressure measurements. Sex, LVMI (binary variable classified by 156 g/m2), and blood pressure were selected as independent variables, and blood pressure was evaluated in each model using VBP, WABP, start of week Before HD-BP, and end of week After HD-BP. For significant models, the p value for each variable and the odds ratio [95% confidence interval] to past CV events were calculated. In model 4, High LVMI and end of week after HD-BP were found to be associated with past CV events. In other words, high LVMI showed an increase in the incidence of past CV events, and low blood pressure after weekend dialysis showed an increase in past CV events (Table 5).
These sentences were added to section 3.5 of the “Results” section.
- Need to incorporate more of the findings in the literature into the discussion
Answer: A recent study that performed sensitivity analysis using LVH and RWT separately showed that LVH but not RWT was associated with higher cardiorenal risk (ref 1). In this study, LVMI and RWT were not significantly different between patients with CV and those without CV events. From the pathophysiologic standpoint, an increase in afterload (i.e., arterial hypertension or an increase in large arteries stiffness) can induce concentric LVH, whereas volume overload (i.e., anemia and hypervolemic states) leads to eccentric LVH (ref 2).
Ref 1: Paoletti, E.; De Nicola, L.; Gabbai, F.B.; Chiodini, P.; Ravera M.; Pieracci, L.; Marre, S.; Cassottana, P.; Lucà, S.; Vettoretti, S.; Borrelli, S.; Conte, G.; Minutolo, R. Associations of left ventricular hypertrophy and geometry with adverse outcomes in patients with CKD and hypertension. Clin. J. Am. Soc. Nephrol. 2016, 11, 271–279.
Ref 2: Colucci, W.S.; Braunwald, E. Pathophysiology of heart failure. In: Heart Disease. A Textbook of Cardiovascular Medicine, 6th Ed.; Braunwald, E.; Zipes, D.P.; Libby, P., Eds.; WB Saunders Company: Philadelphia, PA, USA, 2001; pp. 503–533.
These sentences were added to the “Discussion” section.
Reviewer 2 Report
Overall the paper is well written and presents novel data collection in a population that is notoriously difficult to engage in clinical studies. Although cardiac events are relatively loosely defined in the context of LVMI the number of events makes analysis possible. Interesting admission for fluid overload and pulmonary oedema was not included in as CV event. This is probably the most relevant clinical outcome related to LV function. There should be some explanation as to why this was not included.
The introduction is too long and much of the reference to other papers should be transferred to the conclusion which is too short and fails to compare and contrast the study’s findings with the balance of the published literature.
Table 1 is unnecessary. The other tables should be positioned so that they are on a single page. There seems to be an issues with “LVMI class” in Table 2 the cut-off seems to be 146 not 156 as in the paper and the numbers and percentages are incorrect if I understand correctly. 15 pts with a CV event out of a total of 23 is not 25.8%. The p value seems to suggest LVMI > 146 is protective against CV events.
The data is presented at 6 months for some components and 12 months for others which is confusing. I suggest all data be presented at the end of the study at 12 months. The patients weight and BMI should be included if possible. This data would be available (from BSA measurements) and would make some interesting observations about LVMI and BMI.
The characteristics of those patients with LVMI > 156 should be characterised. This seems to be the group with the highest morbidity but their baseline information is not presented. I would be interested to know how they differed from the balance of the patient group. This could replace Table 1.
The conclusion reproduces the results but fails to really discuss the impact of the results on clinical care and its place in the published literature.
Author Response
The paper was revised according to the suggestions and comments of the reviewer’s comments. In the revised manuscript, revisions are in red and underlined, and deletions are in red with strikeout. Some new comments and changes in the text were added. We sincerely believe that we have adequately answered all queries and issues. We would like to thank the editor and reviewers for the thorough review of our paper, which helped improve the manuscript.
Reviewer’s 2 comments and point-by-point answers
- Interesting admission for fluid overload and pulmonary edema was not included in as CV event. This is probably the most relevant clinical outcome related to LV function. There should be some explanation as to why this was not included.
Answer: Admission for fluid overload and pulmonary edema was included in those with a diagnosis of angina or myocardial infarction. Mere fluid overload due to insufficient dry weight settings was not included in the CV event.
These sentences were added to section 2.3 of the “Materials and Methods” section.
- The introduction is too long and much of the reference to other papers should be transferred to the conclusion which is too short and fails to compare and contrast the study’s findings with the balance of the published literature.
Answer: As you pointed out, we revised the introduction for clarity and brevity.
- Table 1 is unnecessary. The other tables should be positioned so that they are on a single page. There seems to be an issues with “LVMI class” in Table 2 the cut-off seems to be 146 not 156 as in the paper and the numbers and percentages are incorrect if I understand correctly. 15 pts with a CV event out of a total of 23 is not 25.8%. The p value seems to suggest LVMI > 146 is protective against CV events.
Answer: In addition to LVMI as a continuous variable that did not differ significantly between the presence and absence of CV events, we added a categorical variable for LVMI classified by a median of 146 g/m2 that showed a significant difference between groups. However, we believe it is not appropriate to use anything other than the cutoff value (156 g/m2) calculated by the ROC analysis for the presence or absence of CV events. Therefore, the notation 146 g/m2 in LVMI in Table 2 was deleted.
- The data is presented at 6 months for some components and 12 months for others which is confusing. I suggest all data be presented at the end of the study at 12 months. The patients weight and BMI should be included if possible. This data would be available (from BSA measurements) and would make some interesting observations about LVMI and BMI.
Answer: As you pointed out, we unified the presentation of data and used those at 12 months. Some of the actual measurement data were measured 6–12 months after the start of HD, and it was set to 12 months later.
A previous study on the independent effect of BMI reported a greater effect on LVH in women than in men (ref). There are no new discoveries about LVMI and BMI in this study.
Ref: De Simone, G.; Devereux, R.B.; Chinali, M.; Roman, M.J.; Barac, A.; Panza, J.A.; Lee, E.T.; Howard, B.V. Sex differences in obesity-related changes in left ventricular morphology: the Strong Heart Study. J. Hypertens. 2011, 29, 1431–1438.
These sentences were added to the “Discussion” section.
- The characteristics of those patients with LVMI > 156 should be characterised. This seems to be the group with the highest morbidity but their baseline information is not presented. I would be interested to know how they differed from the balance of the patient group. This could replace Table 1.
Answer: Instead of Table 1, we presented the baseline characteristic with the patients grouped according to LVMI <156 g/m2 and LVMI >156 g/m2. We present these in the new Table 3.
- The conclusion reproduces the results but fails to really discuss the impact of the results on clinical care and its place in the published literature.
Answer: We revised parts of the discussion to match the results. We added the following sentence to the “Discussion” section:
With regard to monitoring BP, the time point at which the BP is measured should be defined clearly. Moreover, because the BP of HD patients varies with each HD session as a result of loss of excess fluid, BP fluctuation should be considered. It is important not to limit BP evaluation to only one measurement, such as before dialysis or after dialysis; rather, multiple measurements of BP should be performed and then averaged. In this study, the number of DM cases was significantly higher in patients with CVE than in those without CVE. VBP was significantly higher in patients with LVMI >156 g/m2 than in those with LVMI <156 g/m2. The results of this study indicated that patients with low blood pressure after weekend dialysis showed an increase in past CV events. A previous study reported that systolic blood pressure <110 mmHg and DBP <70 mmHg were independent risk factors of CV events and all-cause mortality (ref), which supports our findings.
Ref: Yamamoto, T.; Nakayama, M.; Miyazaki, M.; Matsushima, M.; Sato, T.; Taguma, Y.; Sato, H.; Ito, S. Relationship between low blood pressure and renal/cardiovascular outcomes in Japanese patients with chronic kidney disease under nephrologist care: the Gonryo study. Hypertens Res. 2011, 34, 1106–1110.